# Dynamics and stage-specificity of between-population gene expression divergence in the *Drosophila melanogaster* larval fat body

Amanda Glaser-Schmitt[ORCID]*, John Parsch

Division of Evolutionary Biology, Faculty of Biology, Ludwig-Maximilians-Universität München, Munich, Germany

* glaser@bio.lmu.de

## Abstract

Gene expression variation is pervasive across all levels of organismal organization, including development. Few studies, however, have examined variation in developmental transcriptional dynamics among populations, or how it contributes to phenotypic divergence. Indeed, the evolution of gene expression dynamics when both the evolutionary and temporal timescale are comparatively short remains relatively uncharacterized. Here, we examined coding and non-coding gene expression in the fat body of an ancestral African and a derived European *Drosophila melanogaster* population across three developmental stages spanning ten hours of larval development. Between populations, expression divergence was largely stage-specific. We detected higher expression variation during the late wandering stage, which may be a general feature of this stage. During this stage, we also detected higher and more extensive lncRNA expression in Europe, suggesting that lncRNA expression may be more important in derived populations. Interestingly, the temporal breadth of protein-coding and lncRNA expression became more restricted in the derived population. Taken together with the signatures of potential local adaptation that we detected at the sequence level in 9–25% of candidate genes (those showing evidence of expression divergence between populations), this finding suggests that gene expression becomes more developmental stage-specific during adaptation to new environments. We further used RNAi to identify several candidate genes that likely contribute to known phenotypic divergence between these populations. Our results shed light on the evolution and dynamics of expression variation over short developmental and evolutionary timescales, and how this variation contributes to population and phenotypic divergence.

## Author summary

Gene regulatory changes are thought to be important when a species adapts to a new environment because they can temporally and spatially fine-tune gene expression. In this study, we compared gene expression across three developmental stages between a population of the fruit fly *Drosophila melanogaster* from its center of origin in Africa and a

**Data Availability Statement:** The RNA-seq data that support the findings of this study are publicly available from Gene Expression Omnibus with the Series identifier GSE218762. The whole-genome

sequence data are publicly available from http://evol.bio.lmu.de/downloads/ for the Dutch population and from the Drosophila Genome Nexus (https://www.johnpool.net/genomes.html) for the Zambian population. All other relevant data are within the manuscript and its Supporting Information files.

**Funding:** This work was supported by a Deutsche Forschungsgemeinschaft (www.dfg.de) SPP1819 Start-up module grant to AGS and the Deutsche Forschungsgemeinschaft grant number 274388701 to JP, which are part of the priority program "SPP 1819: Rapid evolutionary adaptation". The funders had no role in study design, data collection and analysis, decision to publish, or preparation of the manuscript.

**Competing interests:** The authors have declared that no competing interests exist.

derived population in Europe, which has adapted to this new environment. Gene expression differences between the populations were highly specific to each developmental stage. The expression of long non-coding RNA (lncRNA, a type of non-protein coding gene) was higher and more extensive during the last larval stage in Europe, suggesting an increased importance of lncRNA regulation in derived populations. Gene expression became more temporally restricted in the European population, which together with the potential signatures of adaptation we detected in candidate genes (those showing expression changes between populations), suggests that gene expression becomes more temporally-specific during adaptation to new environments. We also found that several candidate genes likely contribute to known trait differences between these populations. Our study helps us further understand the evolution of gene expression variation over short developmental and evolutionary timescales as well as how this variation influences phenotypic differences among individuals.

## Introduction

Gene expression variation can be seen across all levels of organismal organization (i.e. life cycle [1,2], population [3–5], species [6–8], tissue [9,10], and cell [11,12]) and this variation drives phenotypic divergence among species and populations [13,14]. A long-standing goal in population genetics and evolutionary biology has been to identify and characterize this variation, the forces that shape it, and the effects it produces on phenotype. Indeed, understanding the scope and architecture of gene expression variation as well as the mechanisms that drive and maintain phenotypic variation is essential for understanding phenotypic evolution and the maintenance of biodiversity.

Transcriptome studies have been essential in the ongoing characterization of the dynamics and genetic architecture of gene expression variation and its regulation [15–18]. These studies have provided valuable insight into how natural selection shapes gene expression variation and dynamics [8, 19–23]; however, the characterization of the interplay between expression variation, natural selection, and organismal phenotype remains an ongoing effort (reviewed by [24–26]). Studies on the spatiotemporal dynamics of gene expression and its evolution have largely focused on expression variation and divergence across species [18,21–23,27] or within one population or relatively few genotypes of a single species [2,20,28,29]. Indeed, in even the well-studied *Drosophila melanogaster*, temporal expression dynamics have largely been examined within a single population or a limited number of genotypes [2,29,30]. On the other hand, studies on gene expression divergence among *D. melanogaster* populations have largely focused on a single life stage [31–33] or very distant developmental stages [34]. Thus, the evolution of temporal gene expression dynamics when both the evolutionary and developmental timescale are comparatively short remains relatively unexplored. Moreover, these studies have focused almost exclusively on protein coding genes (for exceptions see: [20,23]), missing the potential contribution of non-coding genes, such as long non-coding RNAs (lncRNAs), which have emerged as important regulators of gene expression in the last decade [35] and have been shown to contribute to adaptation [36–38].

Coordinated, and often rapid, changes in gene expression are an integral part of any developmental program and in *D. melanogaster*, transcriptional turnover during development is extensive [1–2]. This is especially true of lncRNAs, which show both tissue- and developmental-specificity, suggesting that they may be of particular importance during development [39, 40], especially at the onset of metamorphosis [40]. During development and its complex

transcriptional program, adult structures are formed and many adult phenotypic traits are determined. Many studies in *Drosophila* have leveraged developmental expression to elucidate the genetic basis and architecture of adult phenotypic traits [22,41–44]; however, these studies have largely focused on comparisons between species. Within *D. melanogaster*, studies cataloguing phenotypic variation within and among populations for developmentally determined adult traits, such as body size and proportion [45–48] and pigmentation [49–52] have primarily focused on measuring a trait in adults and correlating it with genetic variation, which can miss more complex developmental aspects, such as stage- or tissue-specific expression divergence.

In this study, we used high throughput RNA sequencing (RNA-seq) to identify gene expression divergence within and among *D. melanogaster* populations during two late larval and the first prepupal stage in the fat body. This tissue has numerous functions, including energy storage and metabolism, the immune response, detoxification, nutrient sensing, developmental timing, and growth coordination [53–56]. The examined stages span approximately 10 hours of development from the end of feeding behavior to the onset of metamorphosis, and occur surrounding pulses of the maturation hormone ecdysone, which triggers important developmental transitions [53]. Briefly, a pulse of ecdysone triggers the onset of the early wandering stage, when larvae stop feeding and begin wandering until the late wandering stage, during which a large pulse of ecdysone triggers the onset of pupariation, the first stage of which is the white prepupal stage, and metamorphosis. We examined coding and, for the first time to our knowledge, non-coding gene expression divergence between a population from the species' ancestral range in sub-Saharan Africa and a derived, European population, which diverged approximately 12,000 years ago [57], revealing that gene expression dynamics across stages were highly divergent and expression divergence was largely stage-specific. Interestingly, the temporal breadth of gene expression also became more restricted, i.e. more stage-specific, in the derived population. We focused our analysis of non-coding genes on lncRNAs, representing 97% of the non-coding genes that we detected. Expression of lncRNAs was higher and more extensive in the derived population specifically during the late wandering stage, which was the most divergent stage and was also highly variable within another derived, European population. A previous study found that these populations differ in adult body size [58], a trait to which larval fat body expression is known to contribute. Using RNA interference (RNAi), we confirmed a functional role of several novel, differentially expressed genes in body size determination, and likely, body size divergence between these populations. We also found indications of local adaptation, as single nucleotide polymorphism (SNP) divergence was elevated in differentially expressed gene regions.

## Results

In order to quantify gene expression divergence among populations of *D. melanogaster* during development, we performed RNA-seq in the fat body of early and late wandering third instar larvae as well as white prepupae, representing approximately 10 hours of development from the cessation of nutrient intake to the onset of metamorphosis in a European (the Netherlands) and a sub-Saharan African (Zambia) population. In total, 12,454 coding and non-coding genes could be used for analyses; however, it should be noted that not all analyzed genes were detected as expressed in all population and developmental stage combinations (S1 Table). Over 75% of the variation among samples could be explained by stage, while about 10% of the remaining variation could be explained by population (Fig 1A).

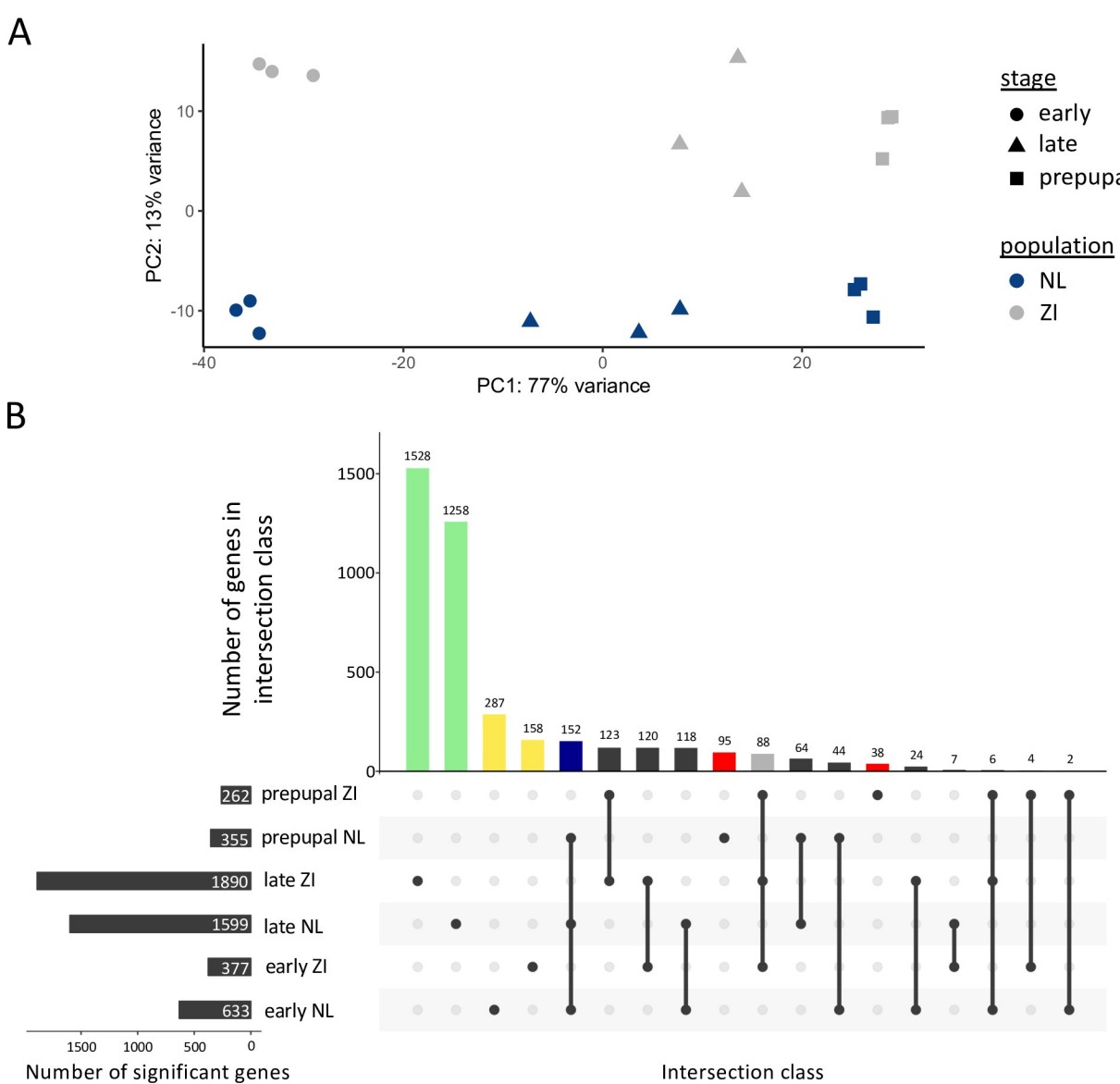

**Fig 1. Interpopulation expression variation between a Dutch (NL) and a Zambian (ZI) population across three developmental stages.** A) Principal component analysis of expression variation in the Dutch (blue) and the Zambian (grey) population during the early (circles) and late (triangles) wandering larvae and prepupal (square) stages. B) Upset plot showing differentially expressed genes and their overlap between the Netherlands and Zambia during the three examined stages. Black, horizontal bars indicate the total number of differentially expressed genes that were significantly up-regulated in one population versus the other population during each stage. A black circle indicates that a stage and population combination is included in an intersection class. Intersection classes with a single circle are comprised of genes significantly up-regulated in a single population and stage. Circles connected by a line indicate intersection classes comprised of multiple stage and population combinations. Vertical bars correspond to the number of genes in each intersection class. Colored, non-black, vertical bars indicate groups of genes up-regulated in a single stage and population (yellow: early; green: late; red: prepupal) or up-regulated across all stages in one of the populations (blue: the Netherlands; grey: Zambia).

## More genes associated with developmental stage than with population

To detect groups of genes with similar patterns of gene expression in the Dutch and Zambian samples, we performed a weighted gene co-expression network analysis (WGCNA), where genes were clustered into modules with correlated gene expression (see Methods). We identified 13 gene co-expression modules, ranging in size from 100 to 2,853 genes (Fig 2A). Next, we

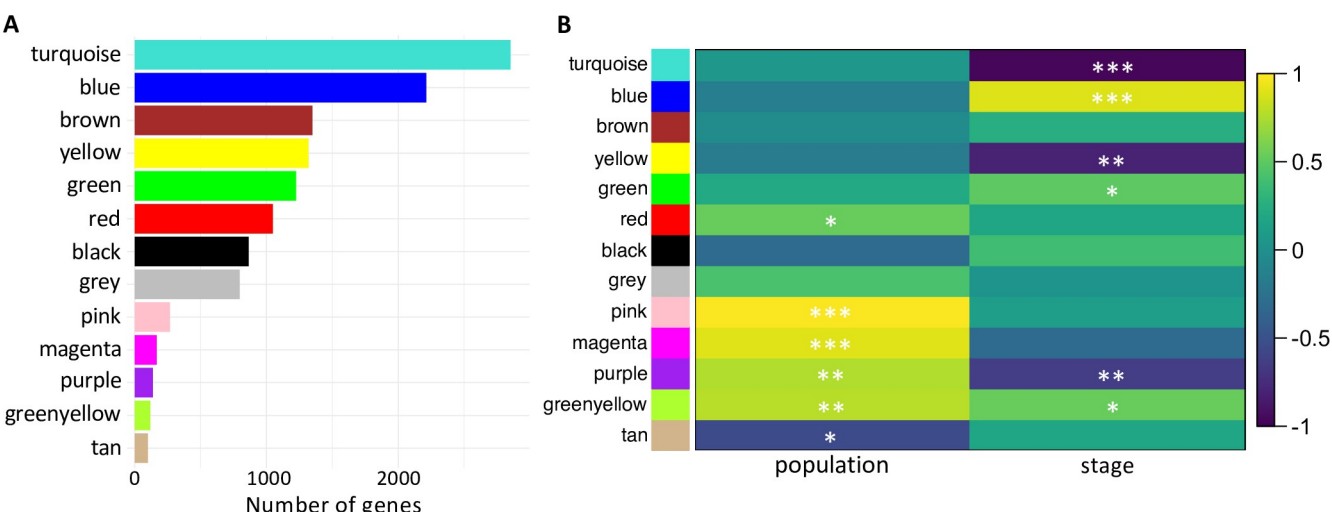

**Fig 2. Results of WGCNA for Dutch and Zambian samples.** A) Size of identified gene co-expression modules. Each module is identified by a color. The grey module contains all genes whose expression did not correlate with any other genes. B) Heatmap of Pearson's correlation between co-expression modules and the traits developmental stage and population. Significant differences between groups were determined using the student asymptotic *P*-value from the Pearson's correlation. * $P < 0.05$, ** $P < 0.005$, *** $P < 10^{-5}$.

correlated module eigengenes, i.e. the module summaries, with developmental stage and population in order to identify modules significantly associated with developmental stage and/or population. Four modules were significantly associated with developmental stage, four with population, and two were associated with both (Fig 2B). More genes were associated with developmental stage than with population, with four of the five largest modules significantly associated with stage (Fig 2). Developmental stage-associated genes were also more highly connected, with the exclusively stage-associated modules being the most highly connected (S2 Table and S1 Data). Approximately 69–81% of genes in modules associated with population were differentially expressed between populations during at least one stage. When we examined the top five most highly connected genes within each module, i.e. the hub genes, at least one hub gene was differentially expressed across all stages for all of the population-associated models, except the tan and red modules, but including all 5 in the pink and purple modules (S2 Table). On the other hand, none of the most highly connected genes in the exclusively stage-associated modules were differentially expressed across all stages (S2 Table) and for three of these modules, only 18.4–27.8% of genes were differentially expressed between populations; while, for a fourth module, the blue module, 46.2% of genes were differentially expressed (S2 Table). Thus, similar to the results of our principal component analysis (Fig 1A), the signal for developmental stage was stronger than that of population despite the large number of genes showing expression divergence.

## Divergent expression between a European and a sub-Saharan African population

Next, we examined genes differentially expressed between the Zambian and Dutch populations during each stage. In total, 4,116 genes, approximately one-third of analyzed genes, were differentially expressed between these populations in at least one stage, with 2,057 up-regulated in at least one stage in the Netherlands and 2,098 genes up-regulated in Zambia. In the early wandering (early), late wandering (late), and white prepupal (prepupal) stages, 1,010, 3,489, and 617 genes, respectively, were differentially expressed (Fig 1B). However, there was very

little overlap in differential gene expression among stages, with the majority of differentially expressed genes being private to one stage (Fig 1B). Indeed, only 88 genes up-regulated in Zambia and 152 genes up-regulated in the Netherlands were shared across all examined stages (Fig 1B). Thus, the majority of differential expression that we detected among the two populations was stage-specific, though the magnitude of expression changes in stages or populations with more differentially expressed genes tended to be lower (S1 Fig). However, when considering the top 20 most divergently expressed genes, i.e., the expression changes detected as the most significant (S3 Table) or those changes with the highest magnitude (S4 Table), 85–100% were shared across at least two stages and 60–95% were shared across all stages (S3 and S4 Tables). Thus, the largest and most consistent gene expression changes were shared across multiple stages.

When we examined the magnitude of expression changes between the two populations, increases in expression were significantly larger in the Zambian than the Dutch population during the early stage but significantly larger in the Dutch population during the late stage, despite these populations having fewer up-regulated genes in their respective stages (S1A Fig; $t$-test, Bonferroni-corrected $P < 10^{-5}$ for both). The magnitude of expression changes was not significantly different between the two populations during the prepupal stage (S1A Fig; Bonferroni-corrected $P = 0.922$). When we compared the magnitude of interpopulation differential expression between stages, the magnitude of expression changes in the late stage, where we detected highest number of differentially expressed genes, was significantly lower than in the early or prepupal stage (S1A Fig; Bonferroni-corrected $P < 10^{-10}$ for both). The magnitude of expression changes in the early stage was higher than in the prepupal stage, but this difference was marginally nonsignificant (S1A Fig; Bonferroni-corrected $P = 0.0518$). Although lowly expressed (S2 Fig) and non-expressed genes (S1 Table) were present among the genes that we detected as differentially expressed between populations, these genes were not overly represented and did not drive the high number of differentially expressed genes that we detected (S2B Fig). Indeed, these differentially expressed genes with very low expression in one population were usually detected as significantly up-regulated in the other population (S2B Fig).

In order to better understand the types of genes up-regulated in these populations, we tested for an enrichment of gene ontology (GO) biological process and molecular function terms, Reactome and Kegg pathways, and InterPro protein domains for genes up-regulated in each population during each stage, as well as across all stages. Similar to expression divergence, the majority of enriched classes were stage-specific (S5 and S6 Tables); however, a general pattern could be seen across the examined stages: Genes up-regulated in the Netherlands tended to be involved in membrane transport, and in particular, metabolism and detoxification. Indeed, genes up-regulated across all stages in the Netherlands were enriched for the pathway terms "ChREBP activates metabolic gene expression" and "fatty acid metabolism" as well as Cytochrome P450 protein domains and two pathway terms related to Cytochrome P450-mediated metabolism (S5 Table). In Zambia, up-regulated genes tended to be related to immunity. For instance, genes up-regulated across all stages in Zambia were only enriched for one term, "immune-induced protein Dim" domains (S6 Table). Interestingly, genes up-regulated in Zambia specifically at the late stage were highly enriched for many GO biological process terms associated with development, growth, and morphogenesis (S6 Table).

## Expression variation within a European population during the late wandering stage

Expression during the late stage was the most divergent, with 1,890 and 1,599 genes up-regulated in Zambia and the Netherlands, respectively, approximately 80% of which was stage-

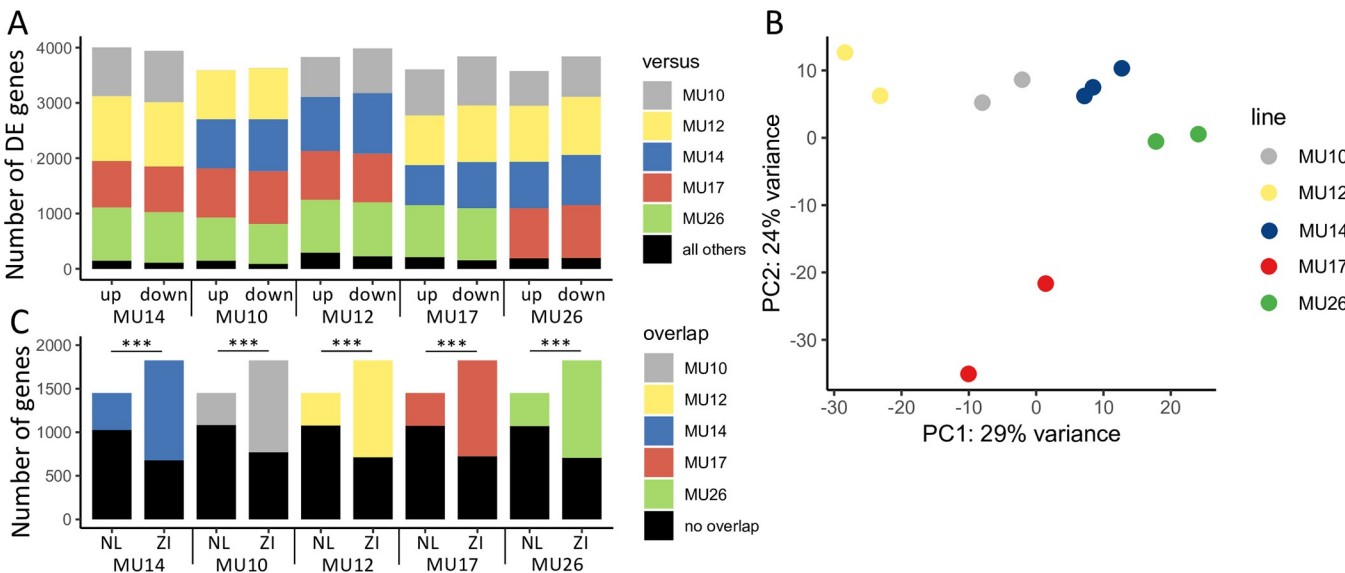

**Fig 3. Expression variation within a German (MU) population during the late stage.** A) Differentially expressed (DE) genes among 5 isofemale strains. Shown are up- and down-regulated genes in a focal strain versus other individual strains and all other strains. B) Principal component analysis of expression variation in the German population. C) Overlap of genes differentially expressed within the German population with genes up-regulated in the Dutch or Zambian populations during the late stage. Differences in the proportions of overlapping genes were determined using a $\chi^2$ test. *** Bonferroni-corrected $P < 10^{-14}$.

specific (Fig 1B). Expression among late samples was also more variable within both populations than in any other stage (Fig 1A). In order to better quantify expression variation within derived populations during this stage, we performed RNA-seq to assay protein-coding gene expression in the fat body of five isofemale strains (rather than pooled population samples) from another derived, European population from Munich, Germany. In total 8,415 genes could be used for differential expression analysis. Approximately 70% of genes (5,817), were differentially expressed between any two strains in this population. Between any pair of strains 1,741–2,598 genes were differentially expressed; while, 237–521 genes were differentially expressed between any single strain and all other strains (Fig 3A). However, despite the large number of differentially expressed genes that we detected in this dataset, variation within strains was relatively low, with biological replicates of each strain clearly clustering together (Fig 3B). Thus, high expression divergence among genotypes and populations appears to be a feature of this stage.

Because we detected so much expression variation within the late stage both within and between populations (Figs 1 and 3), it is possible that many of the genes detected as differentially expressed between the Netherlands and Zambia represent genes with high intrinsic variation that are equally variable within their respective populations, but differ in their representation within the pooled samples that we used for RNA-seq rather than true between-population expression divergence. That is, the genes we detected as differentially expressed may be due to a combination of high inherent variation at the expression level in both examined populations and sampling variation. If this were the case, we would expect genes up-regulated in Zambia to be equally as likely as genes up-regulated in the Netherlands to overlap with genes differentially expressed within Germany. In other words, the genes we detected as differentially expressed between the Netherlands and Zambia should be equally likely to also be differentially expressed in the German population regardless of the population in which they were detected as more highly expressed. However, genes up-regulated in the Netherlands were

significantly less likely to be differentially expressed within Germany than genes up-regulated in Zambia ($\chi^2$ test; $P < 10^{-14}$ for all), suggesting that, in the German population, genes up-regulated in the Netherlands show more consistent expression than genes up-regulated in Zambia. Indeed, the overlap between genes variable within the German population and those up-regulated in the Dutch population was only 31–35% that of the overlap seen with up-regulated Zambian genes (Fig 3C). Thus, the genes we detected as up-regulated in the Netherlands most likely represent consistent expression changes within Europe.

## Divergent lncRNA expression between a European and a sub-Saharan African population

Because ~97% (816 of 839) of the non-coding genes that we detected as expressed in our dataset were lncRNAs, we focused our analysis of non-coding genes on this class. In order to better understand how lncRNAs contribute to expression divergence between populations, we also assayed lncRNA expression in the Zambian and Dutch populations. Eight lncRNAs were among the most divergently expressed genes between these populations (S3 and S4 Tables): four of which were up-regulated in the Netherlands (*CR45823*, *CR44936*, *CR45601*, and *flamenco*) and four up-regulated in Zambia (*CR45045*, *CR44458*, *CR44868*, and *CR43174*). lncRNAs are thought to play an important role in developmental regulation, particularly at the onset of metamorphosis, with many lncRNAs increasing in expression specifically during the late wandering larval stage [40]. Therefore, we examined the relative contribution of lncRNAs to expression divergence between these populations during the examined stages. Interestingly, specifically during the late stage, we found a significant excess of the proportion of lncRNAs up-regulated in the Netherlands, and, at the same time, a significant dearth of the proportion of lncRNAs up-regulated in Zambia (Table 1; $\chi^2$ test; $P < 5 \times 10^{-4}$ for both), suggesting that lncRNAs may make a particularly important contribution to interpopulation expression divergence at this stage. We also found a significant dearth in the proportion of lncRNAs up-regulated in Zambia at the early stage (Table 1; $P = 0.012$). This dearth of up-regulated lncRNAs in Zambia cannot be explained by differences in mapping bias between Dutch and Zambian samples (see S1 Text and S3 Fig), suggesting that lncRNA expression during the wandering larval stages is lower for the ancestral Zambian population in general. Interestingly, when we compared the magnitude of expression changes in genes differentially expressed between populations (Fig 1A), changes in lncRNA genes tended to be larger than for protein coding genes

**Table 1. Relative contribution of lncRNA versus protein coding (PC) genes to genes up-regulated in the Zambian (ZI) and Dutch (NL) populations.**

| Upreg.[a] | Stage | PC genes[b] | lncRNA[b] | $P$[c] |
|---|---|---|---|---|
| NL | Early | 600 | 30 | 0.207 |
| NL | Late | 1451 | 146 | $1.54 \times 10^{-4}$ |
| NL | Prepup | 337 | 17 | 0.224 |
| ZI | Early | 363 | 12 | 0.012 |
| ZI | Late | 1824 | 63 | $7.15 \times 10^{-8}$ |
| ZI | Prepup | 250 | 11 | 0.163 |

[a]Population in which the genes were detected as upregulated.

[b]11,615 protein coding and 816 lncRNA genes were expressed in our dataset.

[c]Significant differences in the proportions of lncRNA and protein coding genes in comparison to the whole dataset were determined using a $\chi^2$ test.

(S1C Fig), which was significant for the Dutch early and late stages as well as the Zambian late stage (*t*-test; Bonferroni-corrected $P < 0.05$ for all).

## Divergent lncRNA expression levels and dynamics between Europe and Africa

In order to more directly compare gene expression levels among all sample types, we calculated relative gene expression in transcripts per million (TPM) for all samples. Dutch lncRNA expression was significantly higher than Zambian expression specifically during the late stage (Fig 4C; *t*-test, $P = 2.1 \times 10^{-10}$). Next, we compared lncRNA expression levels between consecutive stages within the same population, i.e. the dynamics of lncRNA expression within each population. Dutch lncRNA expression increased significantly in the late stage in comparison to the early stage (Fig 3A and 3C; *t*-test, $P = 1.5 \times 10^{-15}$), but did not change significantly between the late and prepupal stages (Fig 3A and 3C; *t*-test, $P = 0.229$). On the other hand, Zambian lncRNA expression did not change significantly between the early and late stages (Fig 3A and 3C; *t*-test, $P = 0.336$), but increased significantly in the prepupal stage in comparison to the late stage (Fig 3A and 3C; *t*-test, $P = 8.9 \times 10^{-5}$). Thus, in general Dutch lncRNA expression increases during the late stage and remains higher during the prepupal stage, while Zambian lncRNA expression remains lower during the early and late stages and then increases later during the prepupal stage (Fig 4A). Protein coding genes showed similar dynamics with a few differences. Expression of protein coding genes was significantly higher in the Dutch population specifically during the late stage (Fig 4D; *t*-test, $P = 1.5 \times 10^{-15}$), but significantly higher in the Zambian population during the early stage (Fig 4D; *t*-test, $P = 2.5 \times 10^{-11}$). When we examined expression dynamics, Dutch protein coding gene expression significantly increased in the late stage (Fig 3B and 3D; *t*-test, $P = 1.5 \times 10^{-15}$) and then significantly decreased in the prepupal stage (Fig 3B and 3D; *t*-test, $P = 0.009$), while Zambian protein coding expression did not significantly increase until the prepupal stage (Fig 3B and 3D; *t*-test, $P = 5.2 \times 10^{-11}$). Thus, we detected shifts in overall protein coding and lncRNA expression levels across the examined stages between these populations.

## Divergent gene expression dynamics between Europe and Africa

To examine how transcriptional turnover differed between Dutch and Zambian populations, we first identified genes differentially expressed among the examined stages within each population and then compared the two populations. Approximately 63% (7,828) of analyzed genes were differentially expressed between two time points in at least one population. Although lowly expressed (S2 Fig) and stage-specific genes (S1 Table) were present among the genes that we detected as differentially expressed between stages within each population, these genes were not overly represented and did not drive the high number of differentially expressed genes that we detected (S2C Fig). Of these genes, 2,019 were exclusively differentially expressed in the Dutch population, while 1,198 were exclusively differentially expressed in the Zambian population. Of the genes differentially expressed between consecutive stages, approximately twice as many were privately differentially expressed or differentially expressed in opposite directions (Table 2). The opposite occurred for differentially expressed genes in non-consecutive stages; however, despite being more conserved, there were still over 2,000 genes with opposite or private differential expression (Table 2). Indeed, when we calculated divergence of transcriptional turnover between populations, both the non-consecutive and consecutive stages were highly diverged between populations, although the consecutive stages were more highly diverged than the non-consecutive (Table 2). However, when we tested for an interaction between population and developmental stage, we only detected 1,036 genes with a

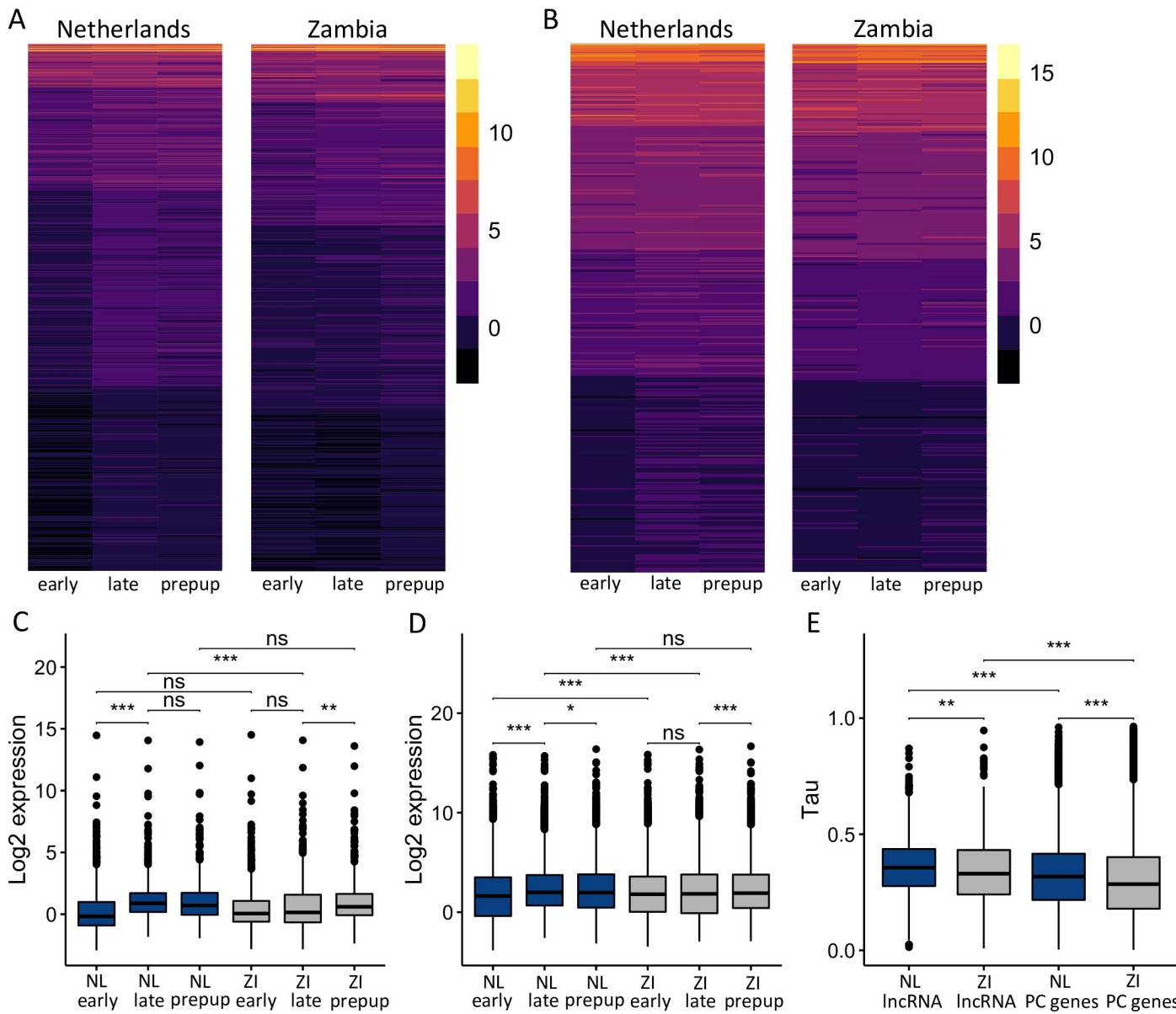

**Fig 4. lncRNA and protein coding gene expression in a Dutch (NL) and a Zambian (ZI) population.** Heatmaps of A) lncRNA and B) protein coding gene log2 expression in a Dutch and Zambian population during the early, late, and prepupal (prepup) stages. Each row represents a gene. Populations were clustered separately based on mean gene expression using k-means clustering (k = 5). Mean C) lncRNA and D) protein coding gene expression for the Netherlands (NL, blue) and Zambia (ZI, grey) during the examined stages. E) Developmental stage specificity, as measured by tau, τ, for lncRNA and protein coding (PC) genes in the Netherlands and Zambia. Significant differences between groups were determined using a *t*-test with a Bonferroni multiple test correction. * $P < 0.05$, ** $P < 0.005$, *** $P < 10^{-5}$. B and D) Because it was not expressed in the NL early stage, *CG16931* was excluded from these analyses.

significant interaction. Moreover, the magnitude of interpopulation expression changes in genes with a significant interaction was significantly smaller than the magnitude of expression changes in other genes detected as differentially expressed between populations (S1B Fig; Bonferroni corrected $P < 10^{-14}$). Thus, the overall developmental program appears to be largely conserved between these two populations. Indeed, despite significant differences in the magnitude of expression changes (S1D–S1F Fig; *t*-test $P < 10^{-14}$ for all), the expression of genes privately differently expressed within one population was significantly correlated with their

**Table 2. Gene expression dynamics in Zambian and Dutch populations.**

| DE Class[a] | Stage Comparison[b] | | |
|---|---|---|---|
| | early vs late | late vs prepupal | early vs. prepupal |
| Conserved | 2522 | 819 | 4122 |
| Private NL | 2389 | 1023 | 1386 |
| Private ZI | 1671 | 772 | 774 |
| Opposite | 58 | 4 | 17 |
| **Divergence[c]** | 0.963 | 1.021 | 0.836 |

[a]Classification of differentially expressed (DE) genes within the Dutch (NL) and Zambian (ZI) populations into similarly DE genes (conserved), genes DE in the opposite direction, or DE genes private to only one population.
[b]Number of genes in each DE class for comparisons between consecutive (early vs. late and late vs. prepupal) and non-consecutive (early vs. prepupal) stages.
[c]Divergence between ZI and NL population transcriptional turnover, as measured by $1 - \rho$ of the log2 fold-change between the examined stages for each population.

expression in the other population (S7 Table; Pearson's correlation, Bonferroni-corrected $P < 0.005$ for all), suggesting that the population-specific differential expression we detected in these genes represents an increase in the magnitude of gene expression differences conserved across the species.

## Developmental specificity increased in the derived population

In insects, lncRNAs have been shown to have higher developmental specificity than protein coding genes, i.e. their temporal distribution is more restricted [40,59]. Therefore, we compared developmental stage specificity of lncRNAs and protein coding genes in Zambia and the Netherlands by calculating tau, τ, for each gene in each population as a measure of stage specificity. Values of τ can range from 0 to 1, with higher values of τ indicating higher specificity. Thus, genes expressed almost exclusively in one stage will have a value close to one, while those expressed similarly across all three stages will have a value closer to zero. Consistent with previous findings [40,60,61], τ was significantly higher for lncRNAs than for protein coding genes in both populations (Fig 4E; t-test, $P < 10^{-9}$ for both). Interestingly, developmental stage specificity for lncRNA and protein coding genes was significantly higher in the Dutch population than in the Zambian population (Fig 4E; t-test, $P < 0.005$ for both), with similar increases in specificity (11.2% and 10.6%, respectively) for both. Thus, we detected a general increase in developmental stage specificity in the derived, Dutch population.

## Population differentiation in genes associated with expression divergence

In order to determine if any of the population divergence between the Netherlands and Zambia that we detected at the expression level could be associated with population differentiation at the sequence level, we calculated whole-genome $F_{ST}$ between these populations and tested for a significant enrichment of highly differentiated (high $F_{ST}$) SNPs in the proximity of genes that were either differentially expressed between the two populations across all examined stages or in a co-expression module associated with population. After multiple test corrections, 9–19% of genes differentially expressed across all stages and 14–25% of genes in population-associated co-expression modules had an excess of highly differentiated SNPs in their whole-gene regions (Table 3), which is consistent with local adaptation occurring at some of these sites. Such an association between expression divergence and linked SNP differentiation

**Table 3. High $F_{ST}$ outlier SNPs in genes differentially expressed at all stages and population-associated modules.**

| Population | Analyzed[a] | Genes[b] | after test correction[c] |
|---|---|---|---|
| ZI | 86 | 15 | 8 |
| NL | 141 | 34 | 27 |
| **Module** | | | |
| pink | 247 | 53 | 37 |
| magenta | 161 | 34 | 24 |
| purple | 131 | 39 | 32 |
| greenyellow | 116 | 19 | 15 |
| red | 1027 | 200 | 138 |
| tan | 100 | 18 | 14 |

[a]Number of genes included in the analysis. Genes on the 4th chromosome or for which we could not identify SNPs were excluded from the analysis.

[b]Number of genes with significantly more high $F_{ST}$ SNPs than expected by chance. Significance was assessed with a $\chi^2$ test.

[c]Number of genes with a significant excess of high $F_{ST}$ SNPs after a Benjamini and Hochberg multiple test correction.

would only be expected in cases of *cis*-regulatory changes between the Netherlands and Zambia [62].

In order to determine if the sequence-level population differentiation we detected is associated with known regulatory regions, we identified experimentally validated regulatory regions in the REDfly database [63] associated with genes differentially expressed across all examined stages in which we detected an excess of highly differentiated SNPs in their gene region (Table 3). We identified 78 known *cis*-regulatory regions associated with 8 genes (1–47 regulatory regions per gene), 73 of which contained highly differentiated SNPs (S8 Table). In order to determine if regulatory regions of genes up-regulated in Zambia or the Netherlands show divergent regulatory motif composition, we tested for an enrichment of regulatory motifs in regulatory regions associated with up-regulation in one population versus those in the other population (see Methods). We identified five motifs, four of which were for transcription factors, as significantly enriched in regulatory regions associated with higher expression in the Zambian population, but did not find any enriched motifs for the Dutch population (S9 Table). Thus, the population differentiation we identified at the sequence and expression level could be associated with known *cis*-regulatory regions as well as changes in predicted transcription factor binding site composition between populations.

## RNAi reveals genes whose expression influences body size

The Dutch population was previously shown to have significantly higher adult body weight than the Zambian population [58]. Genes associated with population differentiation represent candidate genes that may contribute to this phenotypic divergence. It should be noted, however, that because the fat body is a tissue with multiple functions, not all of the candidates are expected to have an effect on body size. Our analyses revealed two genes that are already known to play a role in body size determination: *fezzik* (*fiz*) and *Acetyl-CoA carboxylase* (*ACC*). The first, *fiz*, which has been shown to be a target of positive selection in derived populations [64,65], plays a role in growth rate and wing load divergence between the Dutch and Zambian populations [58]. *ACC* has been shown to play a role in triglyceride storage and glycogen levels [66] as well as thermal body size plasticity [67]. Using RNA interference (RNAi), we tested eight candidate genes to determine if they play a role in body weight, six of which were up-regulated in the Netherlands, *hephaestus* (*heph*), *Cyp9b2*, *Cyp28d1*, *Baldspot*, *Microsomal glutathione S-transferase-like* (*MgSt1*), and *Dachs ligand with SH3s* (*Dlish*), and two

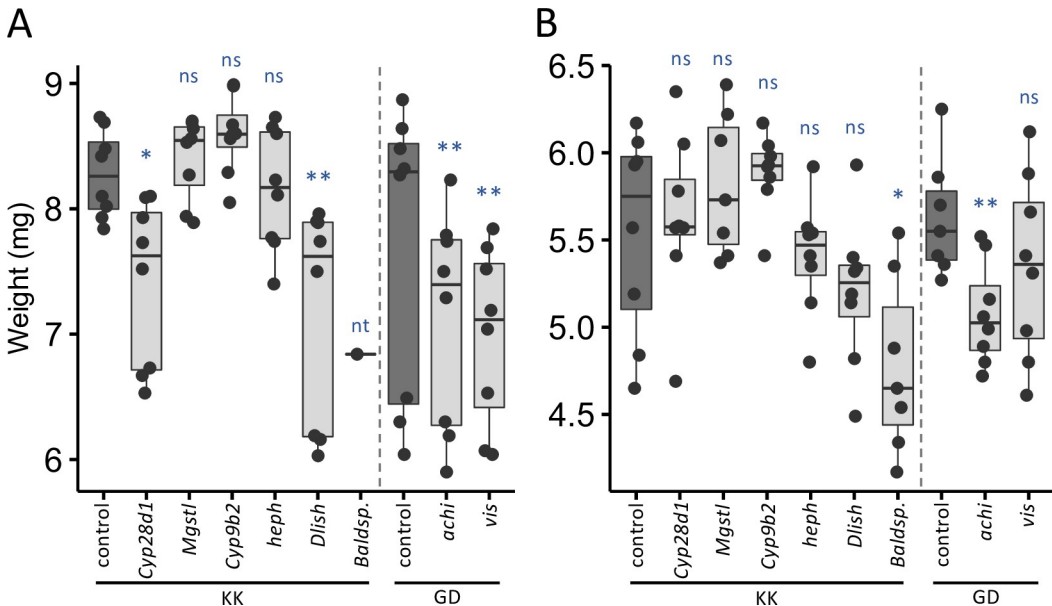

**Fig 5. Body weight in control and RNAi knockdown flies.** Shown are the body weight of groups of five 1–3 day-old female control (dark) or RNAi knockdown (light) flies from the GD and KK Vienna Drosophila Resource Center RNAi libraries [68] on A) standard and B) starvation food. Significance was assessed with an ANOVA. nt not tested, ns not significant, * $P < 0.05$, ** $P < 0.005$, *** $P < 10^{-5}$.

genes up-regulated in Zambia, *achintya* (*achi*) and *vismay* (*vis*). Because genes up-regulated in the Dutch population were enriched for those involved in metabolism, we tested for differences in body size on both standard food and "starvation" food, which contained one-fourth of the nutrients found in standard food. Body weight significantly decreased in comparison to control strains on standard food when we knocked down gene expression of *Cyp28d1*, *Dlish*, *achi*, and *vis* (Fig 5A), and on starvation food when we knocked down expression of *Baldspot* and *achi* (Fig 5B). Body weight of knockdown *Baldspot* flies on standard food was also decreased, but female emergence on the standard diet was very low, and we were only able to perform one replicate (Fig 5A). Thus, from our candidate genes we were able to newly identify five genes that affect body size.

## Discussion

Since their divergence from sub-Saharan African populations ~12,000 years ago [57], cosmopolitan *D. melanogaster* have had to adapt as they expanded into new environments. Using pooled RNA-seq across three developmental stages, representing ~10 hours of development, in the larval fat body of an ancestral African and a derived European population, we found that overall protein-coding and lncRNA gene expression became more developmentally specific (i.e. more temporally restricted) in the derived population (Fig 4). We also detected signatures of potential local adaptation at the sequence level in 9–25% of genes showing evidence of expression divergence between populations (Table 3). This association between expression divergence and linked sequence differentiation is indicative of changes in *cis*-regulation between these populations. Taken together, these findings suggest that gene expression becomes more developmental stage-specific during adaptation to new environments (i.e. the temporal breadth of expression becomes more restricted). However, we should note that because neutral (and nearly-neutral) changes have also accumulated between these

populations, some of the detected changes in stage-specificity may be neutral (or nearly-neutral). In line with this finding, regulatory changes, and *cis*-regulatory changes in particular, are thought to be important during adaptation because they can temporally and spatially fine-tune gene expression [69,70]. Indeed, a recent study found extensive tissue-specific (i.e. spatial) gene expression changes during thermal adaptation [71]. Similarly, we found that gene expression divergence between populations was largely developmental stage-specific (Fig 1).

Previous studies of lncRNA expression revealed that, in comparison to protein-coding genes, lncRNAs are more temporally-specific [40,59], which we also found in this study (Fig 4D). When we examined the contribution of lncRNAs to expression divergence between the examined populations, we found that the relative quantitative contribution of lncRNAs is dependent on developmental stage and population (Table 1). Most notably, lncRNAs were over-represented among the genes up-regulated in Europe and under-represented among the genes up-regulated in Africa during the late wandering stage, suggesting the relative importance of lncRNA expression during this stage is higher for the derived versus the ancestral population. Due to our pooled sequencing approach, it is unclear if this divergence is driven by higher expression in all or a subset of genotypes, an increase in the number of genotypes expressing lncRNAs, or a combination of both. A previous study examining lncRNA expression in a North American population found that several lncRNAs have strain-specific expression, suggesting that lncRNAs can emerge and evolve rapidly [20]. Indeed, in cichlids, the non-coding part of the transcriptome has been shown to evolve more quickly than the coding portion [23].

A previous study using modENCODE transcriptome data found that during larval development, many lncRNAs showed their highest expression specifically during the late wandering stage [40]. We also found that lncRNA expression was significantly higher in the late than in the early wandering stage, but only in the Dutch population (Fig 3A and 3C). In the Zambian population, lncRNA expression was similar between the two larval stages but significantly increased in the prepupal stage (Fig 3A and 3C). The modENCODE data is from a single, derived New World strain, which suggests that increased lncRNA expression during the late wandering stage is actually a derived phenotype. Given the signatures of potential local adaptation we detected (Table 3), this stage-specific increase in lncRNA expression may be adaptive. Indeed, the faster rate of expression evolution in non-coding genes suggests that lncRNAs may play a prominent role in local adaptation, especially over short evolutionary timescales. However, in our dataset, the expression of protein-coding genes in each population showed similar patterns to lncRNA expression (Fig 3A–3D), suggesting that population rather than gene type (i.e. coding versus non-coding) is a more important predictor for overall transcriptome dynamics.

Indeed, we found that gene expression dynamics between the European and African populations were highly divergent (Table 2). However, we could only detect a significant interaction between developmental stage and population in 8% of analyzed genes and many more genes were associated with developmental stage than with population in our WGCNA (Fig 2). These findings suggest that the core developmental program in these populations is still largely intact and that the analyzed stages have a high tolerance for intraspecific expression divergence. The late wandering stage, which coincides with a large pulse of the maturation hormone ecdysone and occurs just before pupariation and the onset of metamorphosis [53], was the most highly divergent between Europe and Africa (Fig 1B). Expression divergence during this stage within another European population was also highly divergent among closely related genotypes (Fig 3), which underscores the importance of the effect of genetic background on expression variation and that even low levels of sequence divergence can lead to large changes in gene expression. This increased expression variation and divergence (Figs 1 and 3) is in line with

the hourglass theory of development, which espouses that morphology, tissue patterning, and gene expression align and are, therefore, most constrained at developmental milestones [72]. Thus, the late wandering stage may be more divergent because it is not a developmental milestone in the fat body. The mechanisms underlying the increased expression variation we detected during the late wandering stage remain unclear but may, for example, be related to developmental variation in transcriptional rates or RNA degradation. It is also possible that the presence of polytene chromosomes in larval fat body cells contributes to expression variation between populations in a stage-specific manner.

Based on candidate genes identified from our analyses, we were able to use RNAi to newly identify five genes that contribute to body weight determination (Fig 5), a phenotype that the fat body is known to affect [53] and has been shown to differ between the examined populations [58]. Body size clines are well-documented in *Drosophila* and are thought to be maintained by selection [45,47,73]. Three of these genes, *Baldspot*, *Cyp28d1*, and *Dlish*, were up-regulated in Europe and likely contribute to body weight divergence between these populations, suggesting that their increase in expression may be adaptive. For *achi* and *vis*, which act in conjunction with one another and were up-regulated in Africa, the effect of RNAi knockdown on body weight was antithetical to expectations if these genes were to contribute to population divergence (Fig 5), suggesting they contribute to body weight variation but not divergence between these two populations. Of these five genes, only *Dlish* has previously been associated with body size variation, specifically wing length variation [74]. Thus, leveraging temporal expression variation and divergence can be a valuable tool in identifying the genes that contribute to phenotypic variation and evolution. Overall, our findings provide insight into the evolution of gene expression specificity during adaption, the evolution of temporal transcriptome dynamics, and how this variation contributes to phenotypic divergence and evolution.

## Materials and methods

### *D. melanogaster* population samples and RNA extraction

All *D. melanogaster* strains were maintained under standard lab conditions (21˚C, 14 hours light: 10 hours dark cycle, cornmeal-molasses media) unless otherwise indicated. RNA-seq was performed for pooled samples of dissected female fat bodies from a European (12 strains from Leiden, the Netherlands) and a sub-Saharan African (10 strains from Siavonga, Zambia) population, as well as individually for dissected fat bodies of five isofemale strains from another European population (Munich, Germany). For the Dutch and Zambian populations, expression was examined during three developmental stages: third instar early wandering larvae, third instar late wandering larvae, and the first white prepupal stage (see S1 Text for more details on staging). For each stage and population, total RNA was extracted from 3 biological replicates, each consisting of a pool of three dissected fat bodies per isofemale strain, for a total of 30 (Zambia) or 36 (the Netherlands) fat bodies per replicate. For each isofemale strain of the German population, total RNA was extracted from 2–3 replicates, each consisting of 15 dissected fat bodies. After larvae were washed in 1X PBS, fat bodies were dissected in cold 1X PBS and stored in RNA/DNA shield (Zymo Research Europe; Freiburg, Germany) at -80˚C until RNA extraction. RNA was extracted with the RNeasy Plus Universal kit (Qiagen; Hilden, Germany) as directed by the manufacturer with the following modification: tissue was homogenized by vortexing.

### RNA sequencing and read mapping

For Dutch and Zambian samples, RNA-seq was performed for 18 RNA-seq libraries. Because this experiment focused on both coding and non-coding genes, ribosomal depletion was

utilized for the isolation of RNA, which was performed along with fragmentation, reverse transcription, library construction, and high- throughput sequencing by Genewiz (Leipzig, Germany). For German samples, RNA-seq was performed for 11 RNA-seq libraries. Because this experiment focused on protein-coding gene expression, poly-A selection was utilized for the isolation of RNA, which was performed along with fragmentation, reverse transcription, library construction, and high- throughput sequencing by Novogene (Hong Kong). For all samples, sequencing was performed using the Illumina HiSeq 2500 platform (Illumina; San Diego, CA) with 150 bp paired reads.

Reads were mapped to the *D. melanogaster* transcriptome (including mRNA, rRNAs, microRNAs, and noncoding RNAs) with the FlyBase release version 6.21 annotation [75] using NextGenMap [76] in paired-end mode. In downstream lncRNA analyses, we included all lncRNAs as defined by FlyBase (517 sense and 219 antisense). Ambiguous read pairs, i.e. those mapping to different genes or transcripts, were discarded. For Dutch and Zambian samples, the total number of reads per library ranged from 35.0–52.6 million, 84.0–91.5% of which could be mapped to protein- and non-coding genes. For German samples, the total number of reads per library ranged from 37.2–53.4 million, with 92.4–98.5% of reads mapping to protein-coding regions. For differential gene expression analyses, we analyzed the sum of read counts across all of a gene's transcripts (across all annotated exons), i.e. on the individual gene-level.

In order to test for potential mapping bias that could be introduced by mapping reads from the ancestral Zambian population, which harbors more genetic variation, to the FlyBase reference annotation (version 6.21 [75]), which is from a lab strain more likely to be closely related to the derived, European populations, we re-mapped reads from the Zambian and Dutch libraries to a Zambian reference transcriptome (strain ZI418), which was created as described in [58]. Briefly, in order to generate the Zambian reference transcriptome, if a nucleotide sequence difference on the major chromosome arms (X, 2R, 2L, 3R, 3L) occurred between the FlyBase annotation and the ZI418 strain, the Zambian variant was substituted into the new reference transcriptome. Due to slight differences in mapping, ~0.27–0.30% of genes could be included in one reference but not the other when the same cutoffs were applied as in the "Differential gene expression analysis" section below. We also restricted our analysis to only the major chromosome arms; therefore, 12,322 genes were included for this analysis. In order to determine if the choice of reference transcriptome introduced any mapping bias, for each sample and gene, we calculated a simple measure of mapping bias: $(n_{zam} - n_{flybase}) / (n_{zam} + n_{flybase})$, where $n_{zam}$ is the number of reads mapped using the Zambian reference and $n_{flybase}$ is the number of reads mapped using the FlyBase reference. This measure of bias should be robust to sequencing depth differences among samples and genes and yields values between –1 (biased towards Zambian reference) and 1 (biased towards FlyBase reference). We did not detect any mapping bias between Dutch and Zambian samples (S3 Fig and S1 Text).

## Differential gene expression analysis

Differential gene expression analysis was carried out using the negative binomial test as implemented in DESeq2 [77] in R [78] with a 5% false discovery rate. Due to differences in sequencing methods and sample composition, German samples were analyzed separately from the Dutch and Zambian samples. In order to prevent the exclusion of biologically relevant genes expressed exclusively in only one stage and/or population or line, for a gene to be considered as expressed in a dataset, we required that the gene have an average of at least 15 mapped reads per sample across all samples of that dataset. Thus, in our pooled RNA-seq dataset with 18 samples, a gene was required to have at least 270 total mapped reads across all samples in order to be considered as expressed, while for our 11 sample German dataset, a gene was required to

have 165 total mapped reads. After the filtering of non-expressed genes, a total of 12,454 protein-coding and non-coding genes were used for differential expression analysis in Dutch and Zambian samples and a total of 8,419 protein-coding genes were used for analysis in the German samples.

Differential expression was detected among Dutch and Zambian samples using a one-factor model design with a grouping variable consisting of six levels representing the six possible combinations of population and developmental stage in order to make direct comparisons between all possible types of samples. Differential expression was detected among German samples using a one-factor model design with five levels corresponding to the five isofemale strains. Removing the German strain with the highest variation (Fig 3) from the analysis did not qualitatively change our findings (S10 Table). In order to identify genes with a significant interaction between developmental stage and population, we implemented a two-factor (stage and population) model design with an interaction term between the two factors and performed a likelihood ratio test of the models with and without the interaction term with a 5% false discovery rate. We tested for differences in the magnitude of gene expression changes among groups using a *t*-test and a Bonferroni multiple test correction to account for running multiple tests.

## Calculation of relative gene expression and developmental stage specificity

In order to directly compare all sample types, we calculated relative gene expression as transcripts per million (TPM) for Dutch and Zambian samples [79]. As described in the "Differential gene expression analysis" section above, a gene was required to have an average of 15 mapped reads across all 18 samples in order to be included in the TPM calculation. Tau was used to calculate developmental stage specificity and was calculated using mean TPM as a measure of gene expression for each gene in each developmental stage (S1 Text). We tested for differences among groups using a *t*-test and a Bonferroni multiple test correction to account for running multiple tests.

## Weighted gene co-expression network analysis

In order to detect correlated patterns among genes in Dutch and Zambian samples, we performed a weighted gene co-expression network analysis (WGCNA) using the WGCNA R package [80] to cluster genes into modules of genes with correlated gene expression. Briefly, variance stabilized normalized gene counts of all analyzed genes were used to calculate a topological overlap (TOM) matrix using a power parameter of $\beta = 6$ for soft-thresholding. Gene co-expression modules were then identified using hierarchical clustering and dynamic tree cutting, which automatically determines the optimal split height for assigning module clusters, with a minimum module size of 30 and a correlation threshold of 0.95 for module merging, based on the high connectivity within the constructed network as well as hierarchical clustering of initial modules (S4 Fig). It should be noted, that this threshold results in no module merging, however, a lower correlation threshold of 0.85 resulted in the merging of only three modules into other, larger modules, two of which, the yellow and green modules, are already very large (Figs 2 and S4). For each gene expression module, the eigengenes, i.e. the module summaries, which are essentially the first principal component for each module, were then correlated with stage age in hours and a binary approximation of population (Dutch = 1 and Zambia = 0) using a Pearson's correlation and a student asymptotic *P*-value to assess significance.

## Gene set enrichment analysis

We used FlyMine [81] to search for an enrichment of gene ontology (GO) biological process and molecular function terms, Reactome and Kegg pathways, and InterPro protein domains for genes up-regulated in each population during each stage as well as across all stages.

## Analysis of sequence-level population differentiation

In order to detect population differentiation at the sequence level, we calculated genome-wide $F_{ST}$ for the Dutch and Zambian populations. Whole-genome sequences are publicly available from http://evol.bio.lmu.de/downloads/ for the Dutch population and from the Drosophila Genome Nexus [82] for the Zambian population. Next, we determined the empirical distribution of $F_{ST}$ between the Dutch and Zambian populations for each of the five major chromosome arms. From these distributions we determined the cutoff above which a SNP is among the top 5% most differentiated SNPs for that chromosome arm. The $F_{ST}$ values for the upper 5% cutoffs for chromosome arms 2L, 2R, 3L, 3R, and X were 0.3237, 0.3608, 0.3756, 0.3401, and 0.3813, respectively. Next, for genes of interest, we calculated the number of SNPs expected to fall into the upper 5% of the respective $F_{ST}$ distribution within the gene region as defined by FlyBase [75]. We then tested for enrichment in the observed versus the expected highly differentiated SNPs within these regions using a $\chi^2$ test with a Benjamini and Hochberg multiple test correction [83] for each group of genes of interest. Known, experimentally validated *cis*-regulatory regions associated with genes containing high $F_{ST}$ SNPs and differentially expressed across all stages were identified using the REDfly database [63]. The AME program [84] in the MEME Suite [85] was used to search for an enrichment of regulatory motifs in the regulatory regions identified for either Zambian or Dutch up-regulated genes using the other population as the control and the JASPAR Core insect database [86] and the combined Drosophila Databases [87–90] and the standard settings.

## Body weight assays

Adult body weight assays were performed for candidate genes in which expression had been knocked down using RNAi constructs under the control of the yeast GAL4/UAS system. *D. melanogaster* strains producing hairpin RNA complementary to *heph* (ID: 110749), *Cyp9b2* (ID: 102496), *Cyp28d1* (ID: 110259), *Baldspot* (ID: 101557), *MgSt1* (ID: 109140), *Dlish* (ID: 110749), *achi* (ID: 49640), or *vis* (ID: 49635) mRNA under the control of a UAS as well as the respective control strains containing an empty vector at the same genomic location (UAS⁻, IDs: 600000, 60100), which we used as a control, were obtained from the Vienna Drosophila Resource Center (Vienna, Austria) [68]. The RNAi and UAS- strains were crossed to a *Cyp6g1*-GAL4 driver strain (*6g1*HR-GAL4-6c), which drives GAL4 expression in the larval fat body, midgut, malpighian tubule, and gastric caecae [91] and was kindly provided by Llewllyn Green. To control for other factors that can contribute to size variation, flies were density controlled similar to in the *D. melanogaster* population samples and RNA extraction section above with 50 first instar larvae in small vials containing either standard cornmeal-molasses medium or "starvation" medium, containing ¼ of the nutrients of standard medium. The wet weight of flies was measured in groups of five 1–3 day-old females, with 1–8 replicates spread over 2 batches per strain. Groups of flies were lightly anesthetized with $CO_2$ and placed in pre-weighed 1.5 mL Eppendorf tubes on ice for 5 minutes before being weighed on a Mettler H51 scale (scale division = 0.01 mg, error = 0.05 mg). The weight of 5 flies was then calculated as the weight of 5 flies and tube minus the weight of the tube. Significance was assessed with an ANOVA.

## Supporting information

**S1 Table. Genes included in our dataset but detected as not expressed in a subset of stages and/or populations.** Shown are the number of genes detected as expressed only in a given subset of stages or populations (pop) but not expressed in the others. For a gene to be considered as expressed in a particular population and stage, we required a minimum of an average of 15 reads for that population/stage combination.
(XLSX)

**S2 Table. Connectivity and differentially expressed (DE) genes in population (pop.)- and age-associated modules.** Shown are the number (num.) of genes in each module; the mean module connectivity (connect.); any significant positive (+) or negative (-) correlations with age or population; genes DE between populations at any stage, across all stages, or number of the 5 most highly connected genes DE between populations during all stages; number of genes DE between stages in any population or both populations and number of the 5 most highly connected genes DE between any two stages; and the number of genes with a significant interaction between developmental stage and population as tested with DESeq2.
(XLSX)

**S3 Table. Top 20 most significant differentially expressed genes between a Zambian and a Dutch population.** Colors represent differentially expressed (DE) genes shared among stages (white: DE in all; blue: DE in 2 stages, orange: unique)
(XLSX)

**S4 Table. Top 20 most differentially expressed genes between a Zambian and a Dutch population.** Colors represent differentially expressed (DE) genes shared among stages (white: DE in all; blue: DE in 2 stages, orange: unique)
(XLSX)

**S5 Table. GO term, pathway, and protein domain enrichment in genes up-regulated in a Dutch population.** Empty cells represent terms with no significantly enriched terms at that stage.
(XLSX)

**S6 Table. GO term, pathway, and protein domain enrichment in genes up-regulated in a Zambian population.** Empty cells represent terms with no significantly enriched terms at that stage.
(XLSX)

**S7 Table. Correlation of gene expression between populations for genes privately differentially expressed between developmental stages in either the Netherlands or Zambia.** Pearson's correlation is shown above the diagonal and the associated Bonferroni-corrected *P*-value is shown below the diagonal.
(XLSX)

**S8 Table. Known *cis*-regulatory elements associated with genes showing strong population differentiation at the sequence and expression level.** Shown are the number of high $F_{ST}$ outlier SNPs in known, experimentally validated *cis*-regulatory elements associated with genes differentially expressed across all examined stages and containing an excess of highly differentiated SNPs in their gene regions.
(XLSX)

**S9 Table. Enriched motifs in regulatory elements associated with genes showing strong population differentiation at the sequence and expression level.** Shown are motifs that were detected as enriched in regulatory elements associated with genes containing an excess of highly differentiated SNPs in their gene regions and up-regulated in either the Netherlands (NL) or Zambia (ZI) across all examined stages.
(XLSX)

**S10 Table. Number of genes detected as differentially expressed in the German population when MU17 is included or excluded in the analysis.** Shown are the number of genes detected as up-regulated (up) in pairwise comparisons of the analyzed German strains (1 and 2) when MU17 is included (with MU17) or excluded (no MU17) in our intrapopulation analysis.
(XLSX)

**S1 Data. Results of gene expression, WGCNA, and mapping bias analyses and body weight in RNAi strains.** A) Results of WGCNA and DESeq2 analyses for Dutch (NL) and Zambian (ZI) populations. Log2 fold change and adjusted *P*-values for within and between population comparisons for early (PS12), late (PS79) and prepupal (WPP) stages as well as adjusted *P*-values from likelihood ratio test (LRT) for an interaction between stage and population are shown. WGCNA module membership is also shown as well as total (kTotal) and within module (kWithin) connectivity. B) Results of DESeq2 analysis for the German (MU) population during the late stage. Log2 fold change and adjusted *P*-values are shown. C) Relative gene expression in TPM for Dutch and Zambian populations. D) Mapping bias in Dutch and Zambian populations. E) Body weight of groups of 5 flies in RNAi and control strains.
(XLSX)

**S1 Fig. Magnitude of detected expression changes.** Shown are absolute values of log2 fold-changes of A) genes detected as differentially expressed between the Netherlands (NL, blue) and Zambia (ZI, grey) within early, late and prepupal stages, B) genes with a significant interaction (sig interact) between population and developmental stage versus genes without a significant interaction (no interact) but detected as differentially expressed (DE) between the Netherlands and Zambia during any stage, C) lncRNA and protein-coding (PC) genes detected as differentially expressed between the Netherlands and Zambia within early, late and prepupal stages, and genes detected as privately differentially expressed (private) within the Netherlands or Zambia between D) early and late stages, E) late and prepupal stages, and F) early and prepupal stages. Significance was assessed with a *t*-test with a Bonferroni multiple test correction. ns not significant, * $P < 0.05$, ** $P < 0.005$, *** $P < 10^{-10}$.
(PDF)

**S2 Fig. Distribution of genes across expression bins.** All analyzed genes were binned in each stage and population combination according to their expression in TPM (very low $\leq 1$, low $\leq 10$, moderate $\leq 25$, high $\leq 50$, very high $>50$). A) All analyzed genes expressed in the Netherlands (NL) or Zambia (ZI) in early, late, and/or pupal stages binned according to their expression level. B) All genes differentially expressed between the Netherlands and Zambia during early, late, or prepupal stages binned according to their expression level. Shown are the number of genes expressed in each stage and population. Blue colors represent genes up-regulated in the Netherlands and grey colors genes up-regulated in Zambia during each respective stage. C) All genes differentially expressed between a given stage and any other stage in either the Dutch or the Zambian population.
(PDF)

**S3 Fig. Distribution of mapping bias.** Dutch (blue) and Zambian (yellow) mapping bias for all genes in A) early, B) late, and C) prepupal samples and for only noncoding genes in D) early, E) late, and F) prepupal samples.
(PDF)

**S4 Fig. Gene co-expression modules from WGCNA.** A) Hierarchical clustering of module eigengenes. Vertical lines indicate alternative correlation thresholds for module merging 0.85 (red), 0.8 (green), and 0.75 (blue). B) Dendrogram of hierarchical clustering for module assignment. Dynamic Tree cut indicates the colored co-expression modules into which clusters were cut. Using a 0.95 correlation threshold for module merging, the merged modules remained the same as the original cut modules.
(PDF)

**S1 Text. Supplementary Methods and Materials.**
(PDF)

## Acknowledgments

We thank Hilde Lainer, Melissa Erika Klug, and Alexandra Reuter for excellent technical assistance in the lab. We thank Llewelyn Green and Phil Batterham for sharing the *Cyp6g1*-GAL4 driver strain with us. We also thank Sonja Grath, Annabella Königer, and the LMU Evolutionary and Functional Genomics group for helpful suggestions and discussion.

## Author Contributions

**Conceptualization:** Amanda Glaser-Schmitt.

**Formal analysis:** Amanda Glaser-Schmitt.

**Funding acquisition:** Amanda Glaser-Schmitt, John Parsch.

**Investigation:** Amanda Glaser-Schmitt.

**Methodology:** Amanda Glaser-Schmitt.

**Project administration:** Amanda Glaser-Schmitt.

**Resources:** John Parsch.

**Supervision:** Amanda Glaser-Schmitt.

**Visualization:** Amanda Glaser-Schmitt.

**Writing – original draft:** Amanda Glaser-Schmitt.

**Writing – review & editing:** Amanda Glaser-Schmitt, John Parsch.

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
