## [Decision Letter · Decision Letter 0]

19 Jan 2023

Dear Dr Glaser-Schmitt,

Thank you very much for submitting your Research Article entitled 'Dynamics and stage-specificity of between-population gene expression divergence in the Drosophila melanogaster larval fat body' to PLOS Genetics.

The manuscript was fully evaluated at the editorial level and by independent peer reviewers.  All three reviewers and the editors are enthusiastic about the manuscript. The dataset is timely, the manuscript is well written, and the analyses are insightful and informative. The combination of genomic analyses and functional gene validations makes the manuscript especially compelling. The reviewers all have suggestions about some of the analyses that we hope will strengthen the paper when addressed.  Based on the reviews, we will not be able to accept this version of the paper, but we are willing to consider a revised version.  We cannot, of course, promise publication at this time.

We therefore ask you to modify the manuscript according to the review recommendations. Your revisions should address the specific points made by each reviewer.

Yours sincerely,

Kelly A. Dyer

Academic Editor

PLOS Genetics

Bret Payseur

Section Editor

PLOS Genetics

Reviewer's Responses to Questions

**Comments to the Authors:**

Reviewer #1: Glaser-Schmitt and Parsch examine gene expression in the larval fat body of two populations of D. melanogaster at three timepoints. The paper is clear and well written and the analyses are well executed. The RNAi experiments are a nice addition to the bioinformatic analyses.

Comments:

Line 556: The inclusion criteria as described does not appear to take into account transcript length, which may bias the inclusion criteria. It looks like this filtering was done prior to the DE analysis. The same issue applies to the calculation of relative gene expression. Also, how often are genes expressed in only one stage or population? How many genes show DE due to lack of expression in one stage or population?

Table 3: Do any of the SNPs identified in the gene regions land in known regulatory elements? Using MEME suite or another program could discover if any of these SNPs are in interesting regions. Additionally you could look for enriched motifs surrounding DE genes.

Is there any difference in the time the Dutch vs. Zambian strains take to go from the early wandering to prepupal stage? This might bear on the differences in lncRNA expression changes between stages for these two populations especially for the late wandering stage since it is not a fat body developmental milestone.

Figure 2B: The MU17 sample seems to be driving a lot of the variance within the German population and is also most different between replicates. Could there be any technical differences between the samples? Does excluding this sample alter your conclusions?

Table 2 - For the genes that show private DE in NL and ZI, do you see a trend in the same direction (LFC) for other population, even if they are non-significant (not DE)?

Line 163 - The statement ‘shared across at least one stage’ is confusing - do you mean shared between at least two stages?

Line 290 - “differentially expressed at some time point in at least one population” - I think you mean between two time points in at least one population.

Figure 4B: can this panel be reordered so that the modules are in the same order as panel A - this would make it easier to quickly compare.

The formatting in tables S1 and S2 needs to be standardized.

Reviewer #2: The Glaser-Schmitt and Parsch manuscript investigates gene expression divergence in the Drosophila melanogaster larval fat body across two geographically distinct populations. The evolutionary theory prediction that natural selection is acting more strongly on life stages preceding reproduction provides a strong argument supporting the examination of spatially varying selection at larval stages.

The manuscript feels already well polished, well organized, well written and is easy to read. The methods are sound, the body of data and analysis is significant and going beyond the genomic patterns to the inclusion of single gene validations is a strong plus to the investigation. I am fascinated by the patterns seen regarding to the non-coding genes and the late wandering stages. Given the quality of the manuscript I do not have many comments, however a few major edits could be made, and I am stating below some more comments and lines of thoughts that, I believe, would strengthen the manuscript.

Major points:

Figure 1B is giving me some trouble. I see counts of differentially expressed genes and counts of genes in intersection classes, but I have trouble to grasp where they come from. The text is not helping as the numbers mentioned are not on the figure. Some class comparisons are based on one dot or two or three dots. May be the authors can describe an example in clear words to guide the reader through the figure, or the figure should be edited and made clearer.

Line 516-522: Although ribo-depletion is a good approach regarding to studying so-called non-coding genes, I don’t think polyA enrichment is redhibitory in studying lncRNAs (the major focus of this paper). Many of the genes annotated as lncRNAs actually do have polyadenylation signals and their transcripts are captured in regular polyA enrichment protocols. The ribo-depletion protocol however provides access to variation in expression of all the snoRNA, asRNA, miRNA etc that are not polyadenylated but also, as far as I know, poorly documented when regarding to evolution of expression variation along latitudinal gradients.

Line 525: “Reads were mapped to the D. melanogaster transcriptome (including mRNA, rRNAs, microRNAs, and noncoding RNAs)”. I think the authors made all the effort to capture expression variation for non-coding genes. However, the paper is focused only on the lncRNA class. This class is interesting, more and more lncRNAs are characterized and found to encode either encode functional micropeptides, or regulate gene expression for example. But it feels a bit like a missed opportunity that nothing is reported about the other classes of non-coding genes. My suggestion is either the manuscript should state early that the manuscript is only targeted toward lncRNAs for the non-coding part and a short justification for why the authors do so. Or, a better option would be a short paragraph reporting if any pattern is seen (or not) for the different classes of non-coding genes like miRNAs, asRNAs for example that are typically closer to expression regulation.

Minor points:

Line 61: There are many papers cited here all under the same topic “all levels of organization”. I think all the papers are relevant here, but a breakdown of the different levels of organization would be giving more perspectives to the reader and clarify what exactly the authors mean.

Given the results highlight strong expression differences between the three larval stages, I think a couple sentences in the introduction reminding the reader about the biological meaning (what the larvae do at each stage) would help strengthening the biological context of this study and fix ideas for the non-specialist readers.

Line 447. I feel like the authors are quite (may be overly?) cautious in their discussion. The discussion seems to circle around the idea that non-coding genes by their fast evolutionary pace are good candidates when examining cases of local adaptation over relatively short evolutionary time, but this is not really stated formally. This somehow echoes the abstract sentence “suggesting that lncRNA expression may be more important in derived populations” where “importance” is a little ambiguous. Given how the authors are making the emphasis on the non-coding part of the genome “We examined coding and, for the first time to our knowledge, non-coding gene expression divergence between a population from the species’ ancestral range in sub-Saharan Africa and a derived, European population” I could imagine such line of ideas being formally (while still being cautious) presented as hypotheses for example.

Larval fat body cells can contain polytene chromosomes. Could that explain the higher variation in expression seen in the late wandering stage?

Reviewer #3: Genetic changes that create evolutionary phenotypic divergence often act by altering where and when given genes are expressed. Most detailed single-gene and functional genomic studies of this phenomenon have primarily focused on gene expression divergence at macroevolutionary timescales, with a fixed developmental stage being examined. In this paper, Glaser-Schmitt and Parsch investigate how gene expression in the fat body changes across three developmental stages in two populations of Drosophila melanogaster (Zambia, Netherlands). The authors provide empirically supported and interesting insights about gene expression variation across fine-scale developmental and evolutionary time: 1) most gene expression variation found across the samples, as expected, is due to developmental stage rather than population identity, 2) gene expression divergence between populations disproportionately happens during certain developmental stages, and 3) regulation of long non-coding RNAs (lncRNAs) is as or more likely than proteins to diverge at this evolutionary timescale. The authors lastly demonstrate that at least one of the phenotypes that have diverged between the populations–body size–is affected when some of the differentially expressed genes are perturbed with RNAi.

I found this to be an interesting study that provides important information about how gene expression varies with claims that are supported by the data. The narrow scope of the experiments–one organ, three time stages, and two populations with known demographic histories–strengthens the manuscript by enabling more precise insights and removing other potentially confounding variables. I have some concerns about plausible artifacts that might be affecting the analyses and some suggestions for improving the clarity of the manuscript, articulated below.

Key Points:

The effect of genes at different expression levels on observed patterns. The authors identify large numbers of differentially expressed genes between developmental stages and between populations throughout the paper. The former is to be expected–changing gene expression is a hallmark of development and differentiation. However, the number of differentially expressed genes in both the former and latter case might be artificially inflated by very lowly expressed genes, which tend to be much noisier in RNA-seq datasets. The authors use a cutoff (15 reads minimum per sample for a gene to be considered) but I think an additional plot showing how differentially expressed genes are distributed across expression bins would both show the extent to which this pattern is driven by or robust to changes in lowly expressed genes. For example, the authors write that the vast majority of the 20 most divergently expressed genes between Zambia and Netherlands populations, as measured by relative-fold change, are shared across developmental stages. This is an important result of the paper, but this pattern could be driven by very lowly expressed genes–for such genes, transcript levels may have increased only marginally in magnitude but largely in proportion, or the genes may be prone to being mismeasured because of technical aspects of the experiment. Aside from strengthening the analysis, having this information would provide information about an important feature of gene expression divergence (e.g, do genes at certain expression levels tend to stay conserved while those at others diverge?) and strengthen the paper overall.

Statistical comparisons and multiple testing corrections. The authors perform numerous pairwise comparisons between developmental stages/populations and use a Bonferroni correction (shown in Fig 3). I am unsure of exactly how the authors calculated the statistics for the highlighted differences/non-differences displayed. I think the t-test + Bonferroni correction is accounting for the number of genes being compared between pairwise samples but the authors are also doing a series of t-tests which should be accounted for with its own multiple-testing correction. If the authors have done this correction already, I would urge them to write it with more detail; otherwise this testing burden should be accounted for.

Effect sizes associated with genetic variants. The authors mention the number of differentially expressed genes that are statistically significant, but the magnitude/effect sizes of the differences are rarely mentioned (the data are presented in the supplementary tables). This information is important for better understanding the types of variants that cause gene-expression divergence and how variants in different developmental stages compare. For example, are there fewer but larger effect size variants between the populations at a certain stage and more but smaller effect size variants at others? Are lncRNAs particularly divergent with respect to magnitude changes? Were the effect-sizes of ~1000 genes with a stage x population interaction (line 301) particularly large? Addressing these questions across the sections the authors see fit is necessary for better understanding the evolutionary history and potential of the regulatory variants and the authors I believe should already have the data from their analyses.

Other comments:

Line 140: The authors state that “...about 10% could be explained by population.” I think they mean 10% of the remaining variation that was unexplained by developmental stage.

Line 163-164: The authors write “However, when considering the top 20 most divergently expressed genes, as measured by the largest fold changes and/or most significant expression changes.”

Lines 207-219: The use of the German populations to polarize the changes between Netherhlands and Zambian populations was a really creative experiment and this paragraph greatly strengthens the authors claims. However, some of the sentences are very confusing and I would urge the authors to clarify them and step by the rationale in a more step-by-step way.

Line 313: Further describing tau is important since this might not be a summary statistic that most readers are familiar with. For example, what would the tau value be for genes expressed (weakly or strongly) in 1, 2, or 3 environments?

Lines 327-349: This paragraph largely describes the numbers resulting from the WGCNA analysis without much biological context for making sense of them–I would provide such context whenever possible. A central takeaway from this paragraph is that a samples’ developmental stage better explains gene expression variation than population, a result also found in the PCA (and recognized as such lines 348-349). To me, this paragraph should be presented immediately after the PCA. Additionally, I think “age” and “stage” are being used interchangeably in this paragraph, which creates some confusion in the text.

Line 407: This paragraph shows important functional data but ends here apruptly. I would urge the authors to summarize the key finding.

Lines 422-427: The language here is a bit imprecise; the detected signatures of potential adaptation (outlier Fst values) are not indicative of changes in cis-regulation between these populations. Rather, there are regions with higher than expected genetic divergence which are also in the proximity of differentially expressed genes, which implicates divergence in cis-acting regulatory sequences. The Fst values associated with variants say nothing about the variants’ mechanism of action; that link is being inferred from other data.

Lines 427: Further, the authors take the increased gene-expression specificity to be adaptive in this section and some of this increase almost certainly is. However, nearly neutral changes are also accumulating between the strains and some of these changes may cause a gene to lose repression during one of the measured stages. Such changes would increase the tau value of a gene, making the gene seem as if it had evolved specificity during adaptation. Because the selection has historically been stronger the in the ancestral, non-bottlenecked population, mutations that are deleterious in Zambia may have evolved as nearly neutral in Netherlands and created such scenarios. This alternative account should be acknowledged.

Figure 5: The authors identify genotypes with statistically significant effects from an ANOVA. I think the correct comparison here is actually an ANOVA followed by Tukey’s HSD comparing the different genotypes to wildtype. I would recommend this approach because the question is explicitly whether the RNAi genotypes are smaller than wildtype, and this method enables such pairwise comparisons. With this approach, the authors should recover all of the significant differences they have identified but may also detect some of the comparisons from the starvation condition.

Overall, I think this is an interesting and clearly presented study that describes how genes diverge in expression during the development of an organ between closely related populations and provides empirical information with which to assess models of gene expression evolution.

**Have all data underlying the figures and results presented in the manuscript been provided?**

Reviewer #1: Yes

Reviewer #2: Yes

Reviewer #3: Yes

PLOS authors have the option to publish the peer review history of their article (what does this mean?). If published, this will include your full peer review and any attached files.

Reviewer #1: No

Reviewer #2: No

Reviewer #3: **Yes: **Mo Siddiq

---

## [Decision Letter · Decision Letter 1]

2 Apr 2023

Dear Dr Glaser-Schmitt,

We are pleased to inform you that your manuscript entitled "Dynamics and stage-specificity of between-population gene expression divergence in the Drosophila melanogaster larval fat body" has been editorially accepted for publication in PLOS Genetics. Congratulations!

Yours sincerely,

Kelly A. Dyer

Academic Editor

PLOS Genetics

Bret Payseur

Section Editor

PLOS Genetics

Comments from the reviewers (if applicable):

Reviewer's Responses to Questions

**Comments to the Authors:**

Reviewer #1: The authors have addressed all of my questions. This is a very nice paper.

Reviewer #2: The authors have addressed all my concerns.

Reviewer #3: The authors have addressed all of my comments and criticisms, and I particularly appreciate the thoughtfulness with which they have clarified some areas of the manuscript.

I have one minor comment, which the authors can address if they see fit.

1. I believe Figure S2B contains all of the data necessary to support the associated claims. However, this is a very dense data figure. A different visualization strategy or splitting up the plot may better serve the authors' goals. Specifically, the thing I believe that makes the panel difficult to digest is that it concurrently is showing 1) genes in different expression categories, 2) which population increased or decreased in expression, 3) the relative magnitude of that change in expression, and 4) the developmental stages at which this is happening.

For example, consider the "early" sub-panel, "ZI" population, 5th column. I interpret the size of the bar to represent the very highly expressed genes in the ZI population that are differentially expressed between ZI and NL populations. Of this ~100 or so differentially expressed genes, half are labeled as "very high ZI (dark gray)" and the other half are "very high NL (dark blue)". Does this mean that among the among highly expressed genes in ZI that are differentially expressed, half of them are even more highly expressed in NL than ZI (dark blue)? There is a lot of information in here, and I think a different visualization strategy (maybe scatter plots with colors for different expression bins, for each stage? or two panels instead of one?) might be another potential way of showing the data that is easier to digest.

As mentioned earlier, all of my major concerns have been addressed in the revision. This paper contributes new, valuable information about how gene expression divergence happens at intermediate evolutionary timescales and how those expression differences are distributed across developmental stages of the organism. This is an elegant study and I appreciate the efforts that have been taken to present the manuscript in a logical, clear way.

**Have all data underlying the figures and results presented in the manuscript been provided?**

Reviewer #1: Yes

Reviewer #2: Yes

Reviewer #3: Yes

PLOS authors have the option to publish the peer review history of their article (what does this mean?). If published, this will include your full peer review and any attached files.

Reviewer #1: No

Reviewer #2: No

Reviewer #3: **Yes: **Mo Siddiq

**Data Deposition**

http://datadryad.org/submit?journalID=pgenetics&manu=PGENETICS-D-22-01344R1

**Press Queries**

---

## [Editor Report · Acceptance letter]

21 Apr 2023

PGENETICS-D-22-01344R1 

Dynamics and stage-specificity of between-population gene expression divergence in the *Drosophila melanogaster* larval fat body 

Dear Dr Glaser-Schmitt, 

We are pleased to inform you that your manuscript entitled "Dynamics and stage-specificity of between-population gene expression divergence in the *Drosophila melanogaster* larval fat body" has been formally accepted for publication in PLOS Genetics! Your manuscript is now with our production department and you will be notified of the publication date in due course.

With kind regards,

Anita Estes

PLOS Genetics

On behalf of:
